# Skin T cells maintain their diversity and functionality in the elderly

Hanako Koguchi-Yoshioka [1,2], Elena Hoffer [3], Stanley Cheuk [3], Yutaka Matsumura [1], Sa Vo[1], Petra Kjellman[4], Lucian Grema[4], Yosuke Ishitsuka[1], Yoshiyuki Nakamura[1], Naoko Okiyama [1], Yasuhiro Fujisawa[1], Manabu Fujimoto[1,2], Liv Eidsmo [3,4,6], Rachael A. Clark[5,6] & Rei Watanabe [1,2,6 ✉]

Recent studies have highlighted that human resident memory T cells ($T_{RM}$) are functionally distinct from circulating T cells. Thus, it can be postulated that skin T cells age differently from blood-circulating T cells. We assessed T-cell density, diversity, and function in individuals of various ages to study the immunologic effects of aging on human skin from two different countries. No decline in the density of T cells was noted with advancing age, and the frequency of epidermal $CD49a^+$ CD8 $T_{RM}$ was increased in elderly individuals regardless of ethnicity. T-cell diversity and antipathogen responses were maintained in the skin of elderly individuals but declined in the blood. Our findings demonstrate that in elderly individuals, skin T cells maintain their density, diversity, and protective cytokine production despite the reduced T-cell diversity and function in blood. Skin resident T cells may represent a long-lived, highly protective reservoir of immunity in elderly people.

[1] Department of Dermatology, Faculty of Medicine, University of Tsukuba, 1-1-1 Tennodai, Tsukuba, Ibaraki 305-8575, Japan. [2] Department of Dermatology, Course of Integrated Medicine, Graduate School of Medicine, Osaka University, 2-2 Yamadaoka, Suita, Osaka 565-0871, Japan. [3] Division of Rheumatology, Department of Medicine Solna, Karolinska Institutet, Visionsgatan 18, L8, 171 76 Solna, Sweden. [4] Diagnostiskt Centrum Hud, Apelbergsgatan 60, 111 37 Stockholm, Sweden. [5] Department of Dermatology, Brigham and Women's Hospital, 75 Francis St, Boston, MA 02115, USA. [6]These authors jointly supervised this work: Liv Eidsmo, Rachael A. Clark, Rei Watanabe. ✉email: rwatanabe@derma.med.osaka-u.ac.jp

T-cell diversity is required for protective immune responses[1]. The diversity of naïve T cells bearing new T-cell receptors decreases with age[2], although the total number of T cells in blood is maintained secondary to a compensatory expansion of memory T cells and longer survival of naïve T cells in older individuals[3–5]. Elderly individuals show decreased anti-pathogen responses[6], and this immune dysfunction is thought to result from decreased T-cell diversity in the peripheral blood[1,7].

Approximately 20 billion T cells reside in the entire human skin[8], and these skin T cells consist of diverse populations of recirculating and resident memory T cells ($T_{RM}$)[9–12]. Mouse infection models have shown that tissue $T_{RM}$ can be effectively developed by local immunization and can provide tissue immunity to pathogens and commensals[13–16]. Accumulation of $T_{RM}$ has been reported in the peripheral tissues of older mice following infection. Researches on human have revealed the similar developmental process and the persistence of pathogen-specific $T_{RM}$ in human skin. $T_{RM}$ specific for herpes simplex virus-2 accumulate in the genital skin and take part in eliminating the infected cells through the rapid production of cytokines at the time of recurrence[17,18]. Skin CD4 T cells reactive to varicella zoster virus persist with maintained function in elderly individuals[19]. Similar to anti-pathogen responses, the development of melanoma-specific $T_{RM}$ is needed for protection against tumor growth[20], and these TRM exert their anti-tumor response via activating dendritic cells and cytotoxic T cells[21]. However, studies on $T_{RM}$ have also revealed the differences between human and murine models leading to the emphasis on the importance of directly evaluating $T_{RM}$ in human specimens[22–24]. Beside the life span and living environment, one of the outstanding differences between human and murine skin immunology is lack of dendritic epidermal

T cells in human skin. dendritic epidermal T cells in murine epidermis competes with CD8 $T_{RM}$ for their survival signals and space[25], thus lack of this large population in human epidermis possibly leads to the release of extra niche for $T_{RM}$.

Here, we demonstrate that skin T cells maintain their density, diversity, and protective cytokine production despite the reduced T-cell diversity and function in blood in elderly individuals, and T-cell profiles are remarkably stable in Japanese and Swedish individuals despite the different environmental and genetic backgrounds of these two ethnic groups.

## Results

**The number and cytokine-producing activity of skin T cells are maintained in elderly individuals.** The density of T cells, both CD4 and CD8 fractions, increased in the epidermis of elderly individuals (Fig. 1a: CD3: $p = 0.0028$, $r = 0.8806$, CD4: $p = 0.0016$, $r = 0.9093$, CD8: $p = 0.0206$, $r = 0.7694$). Since the CD4/CD8 ratio significantly decreased in the epidermis by aging (Fig. 1b: $p = 0.0349$, $r = -0.4736$, Supplementary Fig. 1b: $p = 0.0348$, $r = -0.6484$), it is suggested that CD8 T cells show the stronger tendency of being accumulated in the epidermis. Interestingly, the T-cell density and CD4/CD8 ratio did not change with age in the dermis or the blood (Fig. 1a, b, Supplementary Fig. 1a, b). T cells expressing CD69 and CD103 were enriched in the epidermis regardless of age (Fig. 1c, Supplementary Fig. 1c). However, the frequency of epidermal CD8 T cells expressing CD49a, linked to superior cytotoxic capacity[11], increased with advancing age both in the Japanese and in the Swedish individuals (Fig. 1c: $p = 0.0224$, $r = 0.5075$, Supplementary Fig. 1d: $p = 0.0266$, $r = 0.6728$). CD69$^+$CD103$^+$ dermal CD8 $T_{RM}$ also

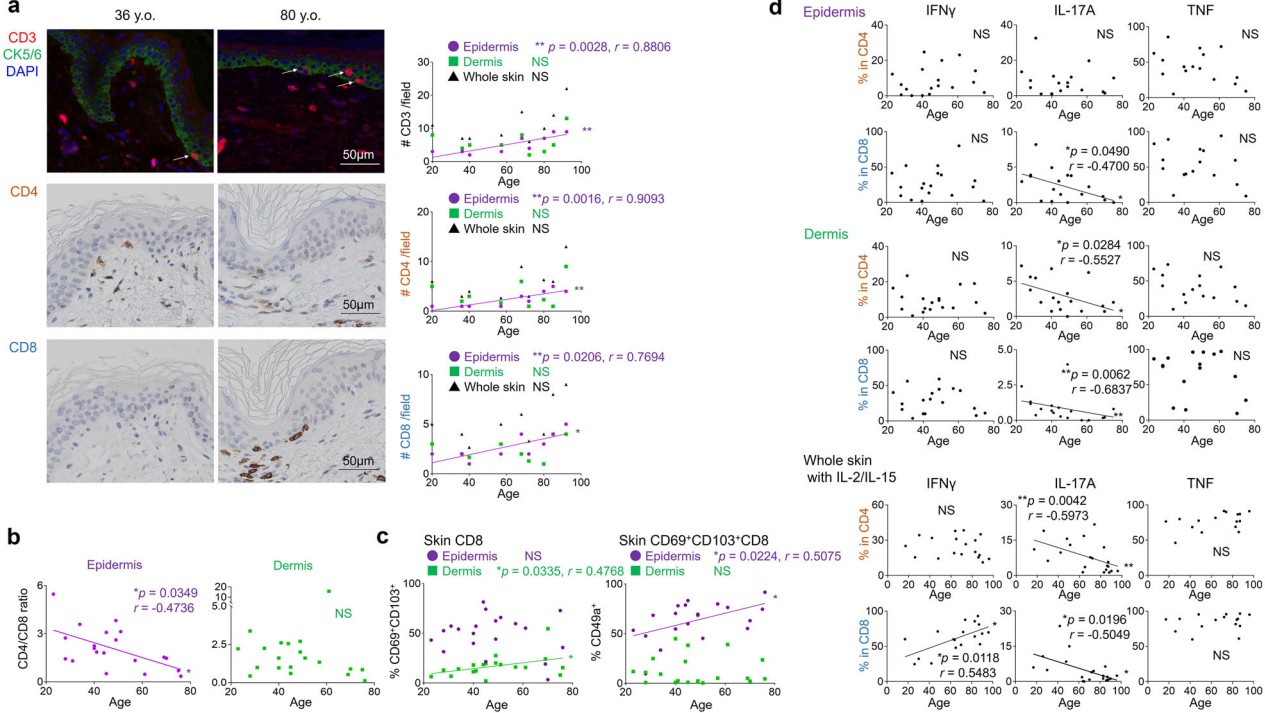

**Fig. 1 The number and cytokine-producing activity of skin T cells are maintained in elderly individuals. a** Representative immunofluorescence (CD3)/immunohistochemistry (CD4, CD8) and CD3$^+$, CD4$^+$, and CD8$^+$ numbers per field (Japanese). In the top immunofluorescence pictures, CD3 (red) and cytokeratin 5/6 (CK5/6, green) are co-stained. Epidermal CD3$^+$ cells are indicated by white arrows. $N = 9$. Bar = 50 μm. **b** CD4/CD8 ratio in epidermis and dermis (Swedish). $N = 20$. **c** CD69/CD103 expression in skin CD8$^+$ and CD49a expression in CD69$^+$CD103$^+$CD8$^+$ (Swedish). $N = 20$. **d** Cytokine production of freshly isolated epidermal T cells, dermal T cells (Swedish), and whole-skin T cells under IL-2/IL-15 (Japanese). Epidermis and dermis (Swedish): IFNγ and IL-17A: $n = 18$, and TNF: $n = 16$; whole skin (Japanese): IFNγ: $n = 18$, IL-17A: $n = 21$, and TNF: $n = 16$. Spearman rank correlation coefficient (two-tailed) was performed in each graph. NS not significant, *$p < 0.05$, **$p < 0.01$.

increased (Fig. 1c: $p = 0.0335$, $r = 0.4768$). The correlation was not observed in the CD4 fractions (Supplementary Fig. 1c, d). The decline of blood naïve T cells was compensated by the expansion of effector memory T-cell ($T_{EM}$) population as previously reported (Supplementary Fig. 1e: CD4 Naïve: $p = 0.0234$, $r = -0.7212$, CD4 $T_{EM}$: $p = 0.0347$, $r = 0.6848$, CD8 Naïve: $p = 0.0088$, $r = -0.7939$, CD8 $T_{EM}$: $p = 0.0149$, $r = 0.7576$)[26,27]. On the other hand, in the skin, in agreement with the increased frequency of skin $T_{RM}$ which demonstrate $T_{EM}$-like phenotype[28], the proportion of recirculating central memory T cell ($T_{CM}$) in skin declined in CD8 fraction of older individuals (Supplementary Fig. 1e: $T_{CM}$: $p = 0.0244$, $r = -0.5490$, $T_{EM}$: $p = 0.0442$, $r = 0.4975$). These results demonstrate a change in the $T_{RM}$ population towards the enrichment of cytotoxic CD49a$^+$ CD8 $T_{RM}$ in the epidermis of elderly individuals. In freshly prepared skin T cells, interferon (IFN) -γ and tumor necrosis factor (TNF) responses were maintained, whilst interleukin (IL) -17A production was significantly decreased in aged skin (Fig. 1d: epidermal CD8: $p = 0.0490$, $r = -0.4700$, dermal CD4: $p = 0.0284$, $r = -0.5527$, dermal CD8: $p = 0.0062$, $r = -0.6837$), although these cytokine productions were comparable in the blood, with constantly low production levels of IL-17A (Supplementary Fig. 1f). To reverse potential T-cell exhaustion in freshly prepared T cells, skin samples were incubated with IL-2/IL-15 for two weeks[8]. Whereas IFNγ production from CD8 T cells increased with age, T cells still showed a significantly reduced production of IL-17A (Fig. 1d: CD8 IFNγ: $p = 0.0118$, $r = 0.5483$, CD4 IL-17A:

$p = 0.0042$, $r = -0.5983$, CD8 IL-17A: $p = 0.0196$, $r = -0.5049$). IL-13 production was not altered by aging (Supplementary Fig. 1g). Our results show that skin T cells maintain IFNγ, TNF, and IL-13 responses whereas IL-17 responses are impaired in aged skin.

**Proliferation and antigen-reactive cytokine production is not impaired in the skin T cells of elderly individuals.** The proliferation of blood and skin T cells in response to the common skin pathogens *Staphylococcus aureus* (*S.au*) and *Candida albicans* (*C.alb*) did not decline with aging (Supplementary Fig. 1h). However, the production of IFNγ and IL-17A was impaired in blood-derived T cells, especially in CD4 fractions, but remained high in skin T cells from elderly individuals (Fig. 2a: *S.au*: blood CD4 IFNγ: $p = 0.0290$, blood CD4 IL-17A: $p = 0.0140$, *C.alb*: blood CD4 IFNγ: $p = 0.0020$, blood CD4 IL-17A: $p = 0.0310$, blood CD8 IL-17A: $p = 0.0290$). In vivo skin responses against tuberculin challenges were also maintained in three Bacille de Calmette et Guérin -vaccinated healthy elderly individuals with negative IFNγ release assay results (Fig. 2b). These data suggest that the antipathogen responses of skin T cells are well sustained in the elderly population.

**The diversity of skin T cells remains high in elderly individuals.** Given that elderly skin T cells display phenotypic and functional changes but maintain antimicrobial responses, we next investigated

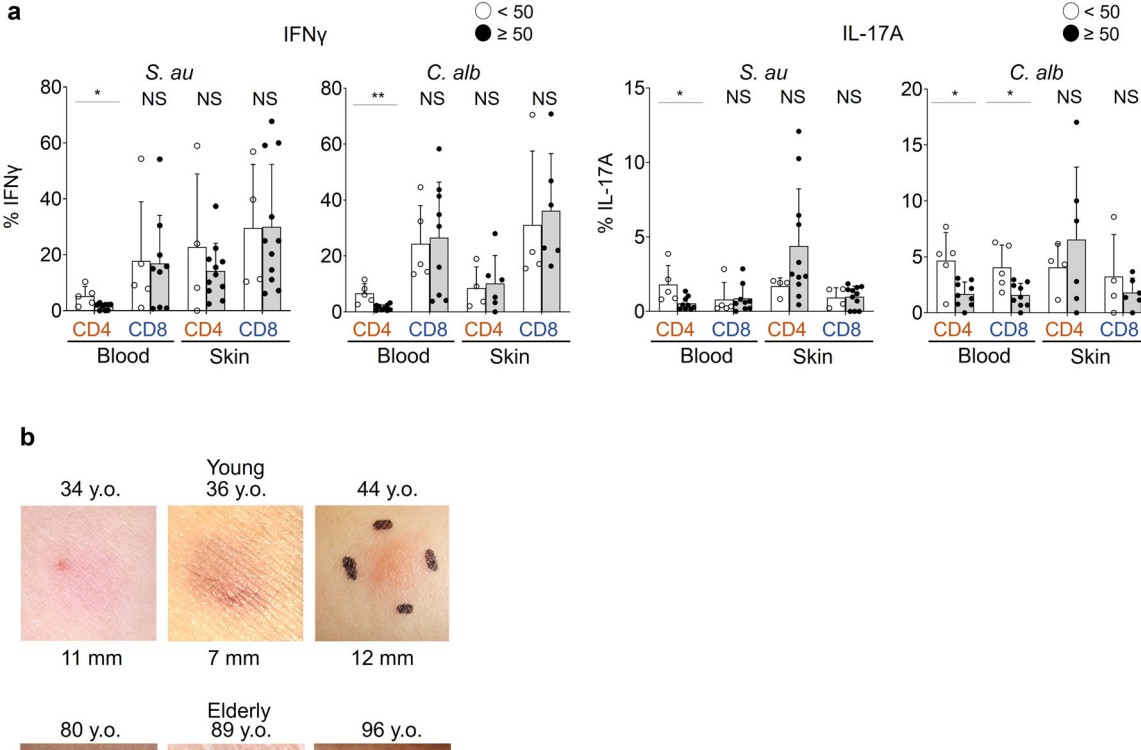

**Fig. 2 Antipathogen responses of T cells decline with age in the blood but not in the skin. a** IFNγ/IL-17A production of blood and skin T cells cultured with heat-killed *S.au* or *C.alb* (Japanese). Error bars indicate standard deviations. Blood: $n = 14$; skin: *S.au*: $n = 15$, *C.alb*: $n = 10$. Mann–Whitney tests (two-tailed) were performed. NS not significant, *$p < 0.05$, **$p < 0.01$. **b** Skin tuberculin reactions of three young and three elderly healthy individuals with previous Bacille de Calmette et Guérin vaccination (Japanese).

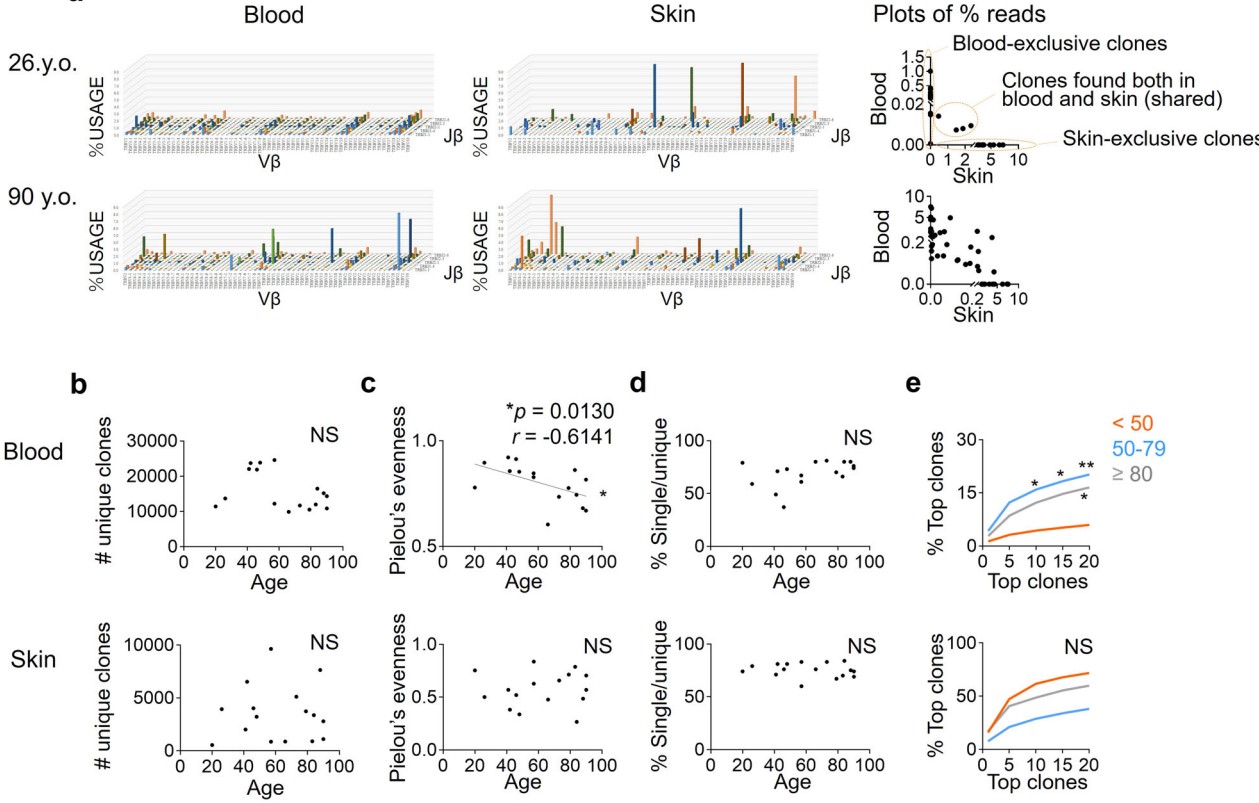

**Fig. 3 T cells from the skin but not blood maintain diversity in older individuals. a** Reperesentative TCRβ repertoires and scatterplots of clone occupancy in the blood and skin. **b** Number of unique clones. **c** Pielou's evenness index (a higher index score means greater diversity). **d** Ratio of single-copy clones in the total unique clones. **e** Mean occupancy of the frequent clones. The cohort was divided into three groups according to age. Samples are from Japanese individuals. N = 16. Spearman rank correlation coefficient (two-tailed) was performed in (**b**–**d**). Two-way ANOVA and Tukey multiple-comparisons tests were applied to (**e**). NS not significant, *p < 0.05, **p < 0.01.

the diversity of T cells in elderly individuals by high-throughput sequencing (HTC) of the complementarity-determining region 3 in T cell receptors (TCRs) from 9π mm² skin and 7 ml of peripheral blood. The number of unique clones, which means the number of distinct TCR clones found in each sample, tended to decrease in the blood of the older individuals, but to be maintained in the skin (Fig. 3a left, 3b). By Pielou's evenness index, elderly skin was presumed to contain diverse T-cell clones whose copy numbers are similar, whilst blood T cells lose diversity with age (Fig. 3c: blood: p = 0.0130, r = −0.6141). Of note, over 60% of skin TCR clones were found as single copies, and this ratio did not decline as the individuals became older (Fig. 3d). The average occupancy of the most frequently detected TCR clones (top 1 clone to top 20 clones) among the total assigned clones was high in the blood but not in the skin of elderly individuals (Fig. 3e: blood: young vs middle: top 10: p = 0.0299, top 15: p = 0.0133, top 20: p = 0.0073, young vs elderly: top 20: p = 0.0320), suggesting that T cells in the blood, but not in the skin, of elderly people become oligoclonal with expansion of limited clones.

**The skin T-cell pool is better maintained than the blood T-cell pool.** Different memory T-cell fractions arise from one naive T-cell clone, and the same TCR clones are found both in the blood and in the skin (shared clones, Fig. 3a right, 4a). In the blood, the ratio of shared clones to blood total unique clones (% shared/unique in the blood) was significantly increased in older individuals, implying that blood-exclusive clones decline with age, whilst the frequency of shared clones in skin total

unique clones (% shared/unique in skin) was comparable throughout aging, implying that skin-exclusive clones live longer than those in the blood (Fig. 4b: blood: p = 0.0065, r = 0.6613). We estimated the relative sizes of the unique clone pool according to the average frequency of shared clones per unique clones. In the participants aged less than 50 years, shared clones occupied 1.32% of the skin T-cell pool and 0.13% of the blood T-cell pool on average. Thus the assumed relative size of the skin T-cell pool vs the blood T-cell pool was 1 vs 10.15 (=1.32/0.13). This ratio became 1 vs 6.42 (=2.12/0.33) in the participants aged 50 to 79 years, and 1 vs 5.09 (=3.41/0.67) in the participants aged 80 years or older. On the basis of this algorithm, the size of the skin T-cell pool was estimated as being 1.99 (=10.15/5.09) times better maintained than that of the blood T-cell pool in elderly individuals (Fig. 4c). These results indicate that T cells remain more diverse in the skin than in the blood of older individuals. Since skin T cells originate from the circulation at some point of life, skin T cells are suggested to persist longer and to form distinct T-cell repertoires in disequilibrium to circulating T cells. Furthermore, when the same clones were sought in different participants (common clones), the number of common clones, which are T cells from different individuals and having identical TCR sequences, increased in the skin but not in the blood of the group aged 80 years or older (Fig. 4d). Thus, T cells with the same TCRs are presumed to be distributed and accumulated in the skin of different individuals. Considering the increase in skin T_RM and the maintenance of antigen reactivity in skin T cells, these T cells found commonly in different elderly individuals are possibly accumulated by recognizing the common antigens in the skin.

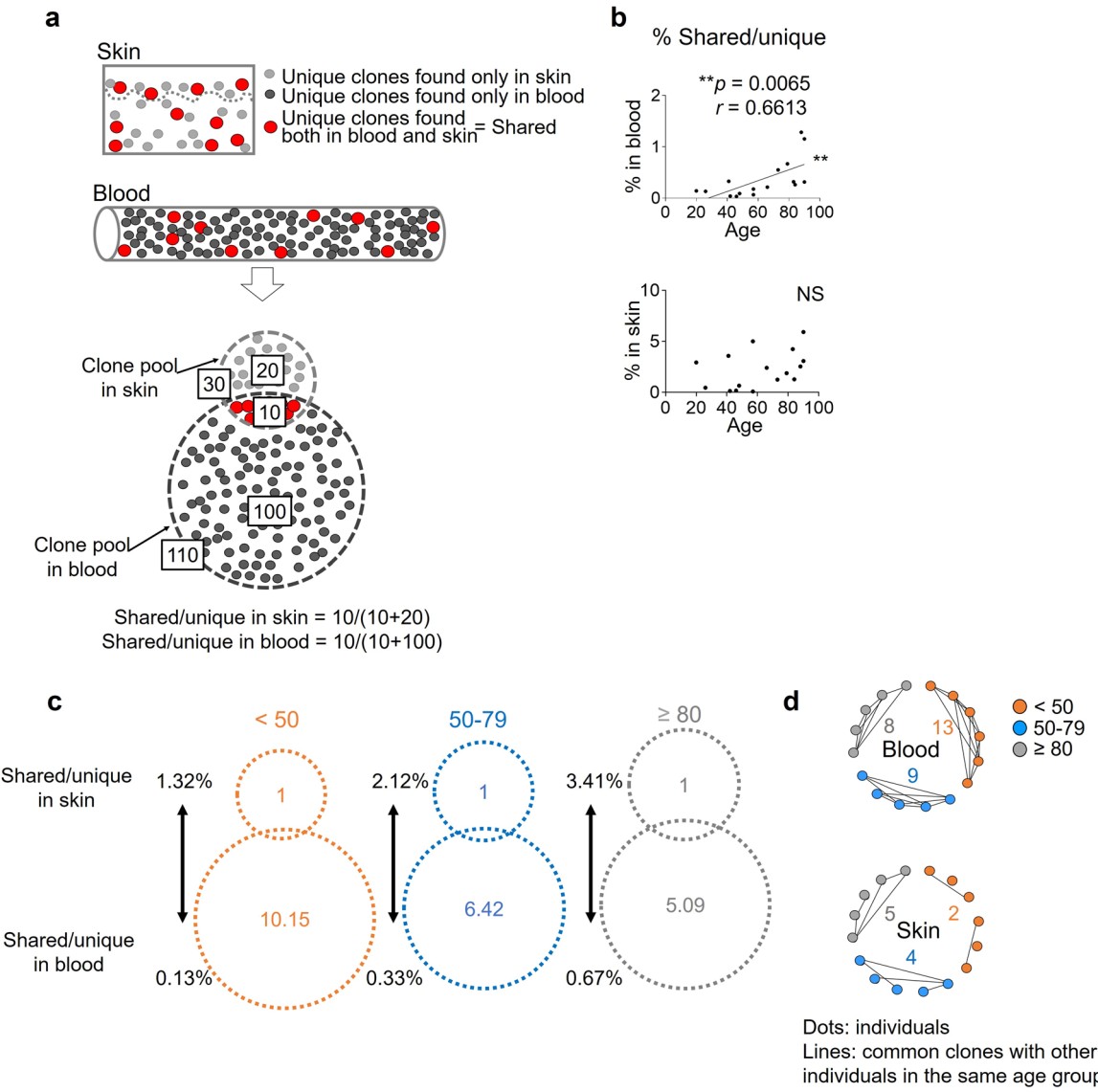

**Fig. 4 The skin T-cell pool is better maintained than the blood T-cell pool. a** Illustration of the unique TCR clones (unique) in the blood (black) and the skin (gray) and the shared TCRs found both in the blood and in the skin (shared, red) of the same individual. In this example, shared/unique is 10/30 in the skin and 10/110 in the blood. **b** Frequency of shared clones per unique clones between the blood and the skin. The two-tailed Spearman rank correlation coefficients were performed. NS not significant, **$p < 0.01$. **c** The relative sizes of the unique clone pools in the skin and the blood were estimated by the average numbers of the shared clones and the % shared/unique in the skin and blood of the 3 age groups. **d** Common clones in the different participants of each age group. NS not significant, **$p < 0.01$.

## Discussion

We found that in elderly individuals, skin T cells maintain their numbers, diversity, cytokine production, and antipathogen responses. Especially, epidermal CD49a$^+$ CD8 $T_{RM}$ increased in elderly individuals in both a Swedish cohort and a Japanese cohort in each independent analysis. Upon unspecific stimulation, IL-17A production from skin T cells was diminished in older individuals with an IFNγ-producing potential in the CD8 T-cell compartment after reversal of cell exhaustion. However, the production of both IL-17A and IFNγ in response to heat-killed *S.au* and *C.alb* was maintained in the skin T cells of elderly individuals. Thus, while skin T cells with IL-17A-producing potential generally decrease through aging, T cells reactive to specific pathogens that are frequent in the skin are presumed to remain in skin sustaining their capacity of producing IL-17A and IFNγ through aging. Expression of CD49a has been linked to IFNγ production and cytotoxicity in skin T cells[11]. An increase in

CD49a$^+$ CD8 $T_{RM}$ in the elderly epidermis may therefore reflect the accumulation of antigen-reactive IFNγ-producing $T_{RM}$ cells over decades of pathogen exposures and increased tumorigenesis through aging.

Skin T cells also maintained a diverse T-cell repertoire into old age, and over 60% of TCR clones in the skin were found as single copies, suggesting that these T-cell clones had not expanded locally. Our findings are consistent with those of studies in animal models that suggest $T_{RM}$ can remain quiescent over a long term in skin in the absence of exogenous antigens[29,30].

We estimated the relative sizes of the T-cell pools in the blood and skin of young versus elderly individuals. On the basis of the occupancy of shared clones between the blood and skin of the same individuals, we could estimate that the skin T-cell pool is twice better maintained than that of the blood T-cell pool of elderly individuals. The increase in shared clones between the blood and skin of the same individuals was significant in the

blood T cells, but not in the skin T cells, suggesting that skin-unique clones also increase concomitantly with aging as skin $T_{RM}$. This result may also imply that blood T cells continuously serve as sources of skin-migrating T cells.

The HTS analysis made it possible to compare the T-cell clones of different individuals. The same T-cell clones were found more frequently within the skin of participants aged 80 years or older than within that of participants aged younger than 50 years, whilst common clones did not increase in elderly blood. Taken together with the results showing the maintenance of proliferation and cytokine production in skin T cells in reaction to pathogens, these results suggest that T-cell clones reacting with specific antigens that the immune system often encounters in skin are preferably accumulated in elderly skin and keep their protective function as long-living $T_{RM}$. Though the mechanism for regulating the maintenance of $T_{RM}$ has not been well defined, one of the possible reasons for the longevity of $T_{RM}$ may be microenvironments. The differentiation fate of $T_{RM}$ depends on the microenvironment of the peripheral tissues[31], and other immune cells such as macrophages and dendritic cells provide niche factors to $T_{RM}$ for their maintenance[32–34]. The lipid-rich microenvironment of skin may also bring out the long-living potential of the adapted skin $T_{RM}$ who are skillful at consuming fatty acids as the energy source in addition to glucose uptake[15,35,36].

This study includes the following limitations. First, the sample size for the HTS analysis was small, and the collected specimens were limited in volume. Only ~1/500,000 of total skin and 1/700 of total blood were analyzed in our study. Thus, most of our findings on the correlation with age were moderate, and we cannot exclude the possibility that our results are biased by an uneven distribution of T cells due to the sites, operating procedures, and timing of the sample collection[37,38], although our subanalyses did not reveal clear differences between the head and trunk (Supplementary Fig. 2a) or between healthy transplant skin and tumor-edge skin in elderly individuals (Supplementary Fig. 2b). Second, we do not have information on the human leukocyte antigens in the HTS analysis. Finally, our analyses were restricted to T cells; the skin microenvironment was not investigated. These limitations should be overcome by further accumulation of human data.

Taken together, our results suggest that local T-cell immunity in the skin is maintained more efficiently in advanced age than is circulating T-cell memory. Our findings may partially explain clinical observations that systemic[39] but not skin infections[40,41] are increased in the elderly. Recent studies have explored novel vaccination strategies to elicit immune memories via local CD8 $T_{RM}$[42–44]. As local restimulation of $T_{RM}$ reportedly leads to the emigration of $T_{RM}$ to lymphoid tissue[45], intracutaneous vaccination in elderly individuals may boost both local and systemic immune memories to previously encountered pathogens such as varicella-zoster virus and *Mycobacterium tuberculosis*. A better understanding of how T-cell responses are maintained in peripheral tissues such as the skin may lead to novel strategies for rejuvenation of systemic immune responses.

## Methods

**Collection of human skin and blood samples, Study approval**. This study was performed on human tissue samples. All the protocols were performed in accordance with the Declaration of Helsinki and were approved by the institutional review board of the ethics committee of the University of Tsukuba Hospital (approval number H28-001) and Karolinska Institutet, the regional ethics committee of Stockholm (reference no. 2012/50-31/2). Written informed consent forms were obtained from all the patients before their inclusion in the study.

A total of 87 skin specimens were obtained from the dermatology department of the University of Tsukuba Hospital as surgical discards of flap reconstruction at least 3 cm apart from the lesion or transplants from patients with benign or in situ malignant tumors. The patients had no internal malignancies or inflammatory diseases other than benign or in situ malignant tumors of the skin. A total of

20 skin samples were obtained from Karolinska University Hospital as surgical discards from the trunk and thighs of healthy patients undergoing reconstructive surgery. Blood from 53 and 17 patients at the University of Tsukuba Hospital and Karolinska University Hospital, respectively, were obtained from the extra preoperation blood tests of the above patients. The details of the samples shown in each figure are provided in Supplementary Table 1. The phenotypic analyses from the Swedish participants (Fig. 1b–d, Supplementary Fig. 1b, c, f) and the TCR repertoire analyses of the Japanese participants (Figs. 3, 4) were conducted from paired samples.

**Isolation of T cells**. Peripheral mononuclear cells were isolated by Ficoll density gradient centrifugation (density 1.077; GE Healthcare Bio-Sciences, IL, USA). As for isolation of epidermal and dermal T cells, whole-skin specimens were incubated in 5 U/ml dispase (Life Technologies, CA, USA) overnight at 4 °C, and the epidermis was separated from the dermis. The epidermis was cut with scissors and incubated in collagenase III (3 mg/ml; Worthington Biochemical Corporation, NJ, USA) for 90 minutes with deoxyribonuclease (5 jig/ml; Sigma-Aldrich, MO, USA) in RPMI 1640 medium with 10% fetal bovine serum. A single-cell suspension was prepared by pipetting. The dermis was digested in the same way in collagenase III with deoxyribonuclease and further processed by a Medicon tissue disruptor (BD Biosciences, CA, USA). Short-term expansion culture was also applied to collect skin T cells from the whole-skin specimens or the epidermal/dermal specimens separated by dispase (Life Technologies) in the presence of 100 IU/ml of IL-2 (Wako, Osaka, Japan) and 20 ng/ml of IL-15 (Wako). The consistency of the T-cell phenotypes isolated from different body sites or different surgical techniques was also investigated in the subanalyses of our results (Supplementary Fig. 2).

**Flow cytometry**. Monoclonal antibodies directly conjugated with fluorescence and isotype controls were used for surface or intracellular staining with optimal concentration for flow cytometry analysis. The antibodies are listed in Supplementary Table 2. Before intracellular cytokine staining, the cells were stimulated with phorbol 12-myristate 13-acetate (50 ng/ml, Wako) and ionomycin (750 ng/ml, Wako) plus Golgi Plug (1 μl/ml, BD Biosciences) for 4–5 h. The cells were surface-stained, fixed, permeabilized, and stained for intracellular and intranuclear targets using BD Cytofix/Cytoperm (BD Biosciences) and True-Nuclear Transcription Factor Buffer Set (BioLegend), respectively, according to the manufacturers' protocols. Analyses of the samples were performed on either a Gallios cytofluorometer, CyAn ADP analyzer (Beckman Coulter, CA, USA) or an LSR-II (BD Biosciences), and the data were analyzed using Kaluza analysis software (Beckman Coulter) or FlowJo (Tree Star, OR, USA). The gating strategies are shown in Supplementary Fig. 3.

**Immunofluorescence and immunohistochemical analysis**. Formalin-fixed paraffin-embedded skin specimens were sliced into 3-μm thicknesses on glass slides. The sliced specimens were deparaffinized and rehydrated, and antigen retrieval was performed using TE buffer (10× concentrate, pH 9.0; Agilent, CA, USA). After they were blocked, as for immunofluorescence, the sliced specimens were incubated with the indicated primary antibodies (CD3 [ab5690, Abcam, Cambridge, UK] and cytokeratin 5/6 [D5/16B4, Agilent]) at a dilution of 1:200 at 4 °C overnight followed by application of secondary antibodies (Alexa Fluor® 555-conjugated donkey anti-rabbit IgG antibody [ab150074, Abcam] and Alexa Fluor® 488-conjugated goat anti-mouse IgG antibody [ab150117, Abcam] at a dilution of 1:1000) for 1 h at room temperature. Mounting medium containing DAPI (VectaShield, Eching, Germany) was used. As for immunohistochemical staining, the sliced specimens were incubated with the antibodies targeting CD4 (4B12, Agilent, 1:50 dilution) and CD8 (C8/144B, Agilent, 1:200 dilution) followed by application of the secondary antibodies (ab97051, Abcam, 1:500 dilution) and diaminobenzidine tetrahydrochloride solution (Vector Laboratories, CA, USA). The sections were counterstained with hematoxylin (Vector Laboratories). Slides were observed with a fluorescence microscope BZ-700 (Keyence, Osaka, Japan). CD3+, CD4+, and CD8+ cells in epidermis, dermis, and whole skin were counted per unit area (300-μm width) of the skin specimens. Three specimens from each study participant were counted, and each set of data shows the average number.

**Proliferation assay**. Peripheral blood mononuclear cells were labeled with 5 μM carboxyfluorescein succinimidyl ester (CFSE) (Cayman, Canada) for 15 min at 37 °C following the manufacturer's protocol. Thirty thousand CFSE-labeled cells were cultured in 10% fetal bovine serum RPMI 1640 medium for a week with or without $10^6$/ml of heat-killed *S.au* (InvivoGen, CA, USA) or $10^6$/ml of heat-killed *C.alb* (InvivoGen, CA, USA). Cell proliferation was assessed by evaluation of CFSE dilution by flow cytometry. Production of IFNγ and IL-17A was also assessed after 4 h of stimulation with phorbol 12-myristate 13-acetate and ionomycin as described above. To measure the proliferation of skin T cells, $9π$ mm² of skin specimen was injected with the above antigens in 250 μl PBS and cultured in 10% fetal bovine serum RPMI 1640 medium for 2 weeks. The T cells were then isolated, and proliferation was evaluated as Ki-67 expression by flow cytometry. IFNγ and IL-17A production was also assessed after 4 h of stimulation with phorbol 12-myristate 13-acetate and ionomycin, as described above.

**HTS of TCR repertoire**. Peripheral blood samples (7 ml) and skin specimens (9π mm$^2$) were collected from 16 participants. Blood RNA was isolated by Isogen (Nippon Gene, Tokyo, Japan), and skin RNA, by RNAlater RNA Stabilization Reagent (QIAGEN GmbH, Hilden, Germany), according to the manufacturers' instructions. The amount and purity of RNA were measured with an Agilent 2100 Bioanalyzer (Agilent). Total RNA was converted to complementary DNA with Superscript III reverse transcriptase (Invitrogen, CA, USA). Then, TCR genes were amplified using an adaptor ligation-mediated polymerase chain reaction. HTS was performed using the Illumina Miseq paired-end platform (2 × 300 bp) (Illumina, CA, USA). Assignment of TRV and TRJ segments in the TCR genes was performed on the basis of the international ImMunoGeneTics information system® (IMGT) database (http://www.imgt.org). Data processing, assignment, and data aggregation were automatically performed using repertoire analysis software (Repertoire Genesis, Osaka, Japan). The amino acid sequences of complementarity-determining region 3 regions between the conserved cysteine at position 104 (Cys104) of the IMGT nomenclature and the conserved phenylalanine at position 118 (Phe118) with the following glycine (Gly119) were translated from the nucleotide sequences. A unique sequence read was defined as a sequence read having no identity in the TRV, TRJ, or deduced amino acid sequence of complementarity-determining region 3 with the other sequence reads. The copy numbers of identical unique sequence read and diversity indexes such as Pielou's evenness were automatically counted using repertoire analysis software in each sample and then ranked according to the copy number.

**Statistics and reproducibility**. Statistical analyses were performed using Prism8 (GraphPad Software, CA, USA). The Spearman rank correlation coefficient was determined using Prism software. Comparison of two groups was performed using the Mann–Whitney test. For the comparison of % occupancy of top clones in the total number of reads in Fig. 3e, data were analyzed using two-way ANOVA and Tukey multiple-comparisons tests. Significant difference was defined as $p < 0.05$ (*), and $p < 0.01$ (**). The exact $p$ values and $r$ values are listed in Supplementary Table 3. Independently analyzed Japanese and Swedish cohorts are included in this study in order to assure the reproducibility of our findings.

**Reporting summary**. Further information on research design is available in the Nature Research Reporting Summary linked to this article.

## Data availability
The authors declare that data supporting the findings of this study are available within the paper and its supplementary information files. Raw data for the graphs can be found in the Supplementary Data file. Additional raw data generated during and/or analyzed during the current study are available from the corresponding author upon reasonable request.

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

## Acknowledgements

This work was supported by the Japanese Association of Geriatric Dermatology Research (to H.K-Y.), JSPS kakenhi grant number JP18K08291, a JSID fellowship Shiseido Research Grant (to R.W.), a Wallenberg Clinical Fellowship, and the Swedish Research Council and Ragnar Söderberg Stiftelse (to L.E. and E.H.).

## Author contributions

H.K-Y.—designed the study, wrote the draft of the manuscript, and contributes to the data collection and interpretation. E.H.—designed the study and collected and interpreted the data. S.C., Y.M., S.V., P.K., L.G., Y.I., Y.N., N.O., and Y.F.—contributed to the data collection and interpretation. M.F. and R.A.C.—analyzed and interpreted the data and assisted in the preparation of the manuscript. L.E., and R.W.—designed the study, analyzed and interpreted the data, and assisted in the preparation of the manuscript. All the authors critically reviewed the manuscript, approved the final version of the manuscript, and agree that questions related to the accuracy or integrity of any part of the work have been appropriately investigated and resolved.

## Competing interests

The authors declare no competing interests.
