## [Peer Review File · Communications Biology]

Reviewers' comments:

Reviewer #1 (Remarks to the Author):

The study by Koguchi-Yoshioka¹ et al is a very interesting piece of work, addressing the increasingly timely issue to understand more about the effects of aging on human immune functions.

They systematically and carefully investigated skin T-cell density, diversity, and function in over 50 individuals of various ages from sampled in Sweden and Japan. They show that in elderly individuals, skin but not blood T cells maintain their density, diversity, and protective cytokine production.

The manuscript is well written and study is well designed and robust. Authors are mindful of certain technical limitations and do not overinterpret their findings.

Minor points

1. Pie charts and statistical comparisons don't work well together, please express data in figure 1D in a different way to highlight statistically significant differences.

2. Colour coding should be consistent and help understanding not confuse. As blue and orange are used to denote epidermis and dermis in Fig 1A, it is confusing that the same colours are then used in Fig 1D for CD4 and CD8 as well.

3. Change TNF- α into TNF according to current nomenclature.

Reviewer #2 (Remarks to the Author):

Dear authors, I have been reading your recent work on the human skin T cell population and its population consistency over age. I found it very interesting and you have managed to approach the study in a good way, which made it easy for me to read. (1) It might be good to add some more explanatory sentences on human skin immunology and what the T-cells do there, how many they are and explain more into why it's important to look at human skin and not murine skin. I like the fact that you have not only concentrated on one population or geographical setup. Per se the skin is the organ that is constantly in contact with the outer environment and that will also be a factor to consider in any study referring to skin immunology or response.

The limitations of the study are completely natural for a study based on human specimens and even though there might be fluctuations in the results if performing similar studies in other groups in the future the overall message, that human skin has "less" age-dependent T-cell population, should sustain.

(2) As a skin biologist I would have appreciated to have normal histology pictures of the skin samples taken and that all pictures, where skin is shown, could be a bit bigger. Also, a staining of for example colVII or a similar BM related protein would be of great help for non-skin experts to see where the dermis ends and the epidermis begins. Also, by making small arrows to point it out.

I think this research is very relevant for our present SARS-Cov-19 pandemic due to its relation to human T-cell residency and prolonged effects and hope to see more work from this group/s concerning this important population of cells regarding viral components.

In conclusion, I enjoyed reading and reviewing your manuscript and with the few changes (marked with numbers in the text) or suggestions given, you will be able to reach a wider audience.

Wishing you success in your future endeavors.

Reviewer #3 (Remarks to the Author):

The manuscript by Yoshioka et al., reported that Skin T cells have different phenotypic and functional properties from those in the blood of elderly people. They found that skin T cells maintain their density, diversity, and protective cytokine production in elderly people. These findings provide a useful source of data for understanding skin T cells. There are several concerns which need to be addressed.

1. In Fig. 1A,S1B, the authors found that the CD4/CD8 ratio decreased in the epidermis of elderly individuals, To analyze the changes of CD4 and CD8 cell numbers, the authors should examine the density or cell numbers of CD4 and CD8 in the epidermis.
2. The authors should examine T effector cells, Tem, Tcm along with Trm to obtain a more comprehensive understanding of a dynamic change in populations of T cells with advancing age.
3. IFN-g and TNF-a production from CD8 T cells was significantly enhanced in the epidermis of elderly individuals in Fig. 1D, which may be due to the increased CD49a+ CD8 Trm cells (Fig. 1C), not the shift from TC17 to TC1.
4. Fig. 2A shown that the production of IFN γ and IL-17A induced by pathogens (*S.au* and *C.alb*) was comparable in CD8 cells from young and elderly individuals, however, IL-17A production from CD8 T cells was significantly decreased in aged skin in Fig.1D. The authors should discuss it.
5. In Fig. 4, skin T cells live longer than those in blood, what is the main mechanism for long-lived CD8 T cells in skin. The author should discuss it.

<Point-by-point responses to the reviewers' comments>

COMMSBIO-20-1609-T: Skin T cells in older individuals remain diverse and functional

We largely appreciate all the reviewers' instructive comments and suggestions. We tried our best to revise our manuscript according to all the reviewers' comments. During the revision, we realized that we incorrectly described the numbers of the subjects analyzed in Figure 1D and Supplementary Figure S1F (Swedish cohort). We sincerely apologize for this careless mistake. The actually included data are not changed. Our point-by-point responses are as follows. Thank you very much again for your kind consideration.

Reviewers' comments are Italics.

Reviewer #1 (Remarks to the Author):

The study by Koguchi-Yoshioka et al is a very interesting piece of work, addressing the increasingly timely issue to understand more about to effects of aging on human immune functions.

They systematically and carefully investigated skin T-cell density, diversity, and function in over 50 individuals of various ages from sampled in Sweden and Japan. They show that in elderly individuals, skin but not blood T cells maintain their density, diversity, and protective cytokine production. The manuscript is well written and study is well designed and robust. Authors are mindful of cetian technical limitations and do not overinterpret their findings.

Minor points

1. Pie charts and statistical comparisons don't work well together, please express data in figure 1D in a different way to highlight statistically significant differences.

→ We appreciate your precious comments and apologize for the inadequate way to show the data. We changed the pie charts in Figure 1D and Supplementary Figure S1F to the dot plots in order to correctly reflect our findings. Though the *p* values are changed due to the different statistical processing, our interpretation of the results are not affected.

D Directly isolated T cells

Swedish cohort

Epidermis

Dermis

Japanese cohort

Whole skin with IL-2/IL-15

F Swedish cohort

Blood

2. Colour coding should be consistent and help understanding not confuse. As blue and orange are used to denote epidermis and dermis in Fig 1A, it is confusing that the same colours are then used in Fig1D for CD4 and CD8 as well.

→ We appreciate your comments and apologize for the confusing color labels. We unified the denoting colors as follows throughout the revised figures; Epidermis: violet, Dermis: green, CD4: orange, and CD8: blue.

3. Change TNF- α into TNF according to current nomenclature.

→ We appreciate your point and changed the descriptions from TNF α to TNF.

Reviewer #2 (Remarks to the Author):

Dear authors, I have been reading your recent work on the human skin T cell population and its population consistency over age. I found it very interesting and you have managed to approach the study in a good way, which made it easy for me to read.

(1) It might be good to add some more explanatory sentences on human skin immunology and what the T-cells do there, how many they are and explain more into why it's important to look at human skin and not murine skin.

→ We appreciate your precious comment and added the following sentence to the introduction part regarding the known human skin immunology and the difference between human skin and murine skin (p.5, l.9~p.6, l.6).

‘Approximately 20 billion T cells reside in the entire human skin, and these skin T cells consist of diverse populations of recirculating and resident memory T cells (T_{RM}). Mouse infection models have shown that tissue T_{RM} can be effectively developed by local immunization and can provide tissue immunity to pathogens and commensals. Accumulation of T_{RM} has been reported in the peripheral tissues of older mice following infection. Researches on human have revealed the similar developmental process and the persistence of pathogen-specific T_{RM} in human skin. T_{RM} specific for herpes simplex virus-2 accumulate in the genital skin and take part in eliminating the infected cells through the rapid production of cytokines at the time of recurrence. Skin CD4 T cells reactive to varicella zoster virus persist with maintained function in elderly individuals. Similar as anti-pathogen responses, the development of melanoma-specific T_{RM} is needed for protection against tumor growth, and these T_{RM} exert their anti-tumor

response via activating dendritic cells and cytotoxic T cells. However, studies on T_{RM} have also revealed the differences between human and murine models leading to the emphasis on the importance of directly evaluating T_{RM} in human specimens. Beside the life span and living environment, one of the outstanding differences between human and murine skin immunology is lack of dendritic epidermal T cells (DETCs) in human skin. DETCs in murine epidermis competes with CD8 T_{RM} for their survival signals and space, thus lack of this large population in human epidermis possibly leads to the release of extra niche for T_{RM}'.

I like the fact that you have not only concentrated on one population or geographical setup. Per se the skin is the organ that is constantly in contact with the outer environment and that will also be a factor to consider in any study referring to skin immunology or response.

The limitations of the study are completely natural for a study based on human specimens and even though there might be fluctuations in the results if performing similar studies in other groups in the future the overall message, that human skin has "less" age-dependent T-cell population, should sustain.

→ We appreciate your comments and evaluating our message.

(2) As a skin biologist I would have appreciated to have normal histology pictures of the skin samples taken and that all pictures, where skin is shown, could be a bit bigger. Also, a staining of for example colVII or a similar BM related protein would be of great help for non-skin experts to see where the dermis ends and the epidermis begins. Also, by making small arrows to point it out.

→ We appreciate and totally agree with your comment. We changed the pictures in Figure 1A to the bigger ones stained for CD3 (red) and cytokeratin 5/6 (green) to clarify the epidermal basal cell layer and epidermal-dermal junction. White arrows indicate epidermal CD3⁺ cells. These pictures help us recognize that T cells in epidermis are mainly in the basal cell layer. We also added immunohistochemical pictures of CD4 and CD8 in Figure 1A, where the epidermal-dermal junction is more obvious, according to another reviewer's suggestion.

I think this research is very relevant for our present SARS-Cov-19 pandemic due to its relation to human T-cell residency and prolonged effects and hope to see more work from this group/s concerning this important population of cells regarding viral components.

In conclusion, I enjoyed reading and reviewing your manuscript and with the few changes (marked with numbers in the text) or suggestions given, you will be able to reach a wider audience.

Wishing you success in your future endeavors.

→ We largely appreciate your thoughtful comments. We believe that further elucidating the mechanisms of anti-pathogen T_{RM} formation and their maintenance will be of help for establishing prevention strategy for infection including pandemic virus. We also made the images in the original Figure 2 bigger according to your comment in the file on the original Figure 2B, ‘Please make these images a bit bigger and tell the audience what it is you would like us to see’.

Reviewer #3 (Remarks to the Author):

The manuscript by Yoshioka et al., reported that Skin T cells have different phenotypic and functional properties from those in the blood of elderly people. They found that skin T cells maintain their density, diversity, and protective cytokine production in elderly people. These findings provide a useful source of data for understanding skin T cells. There are several concerns which need to be addressed.

1. In Fig. 1A, S1B, the authors found that the CD4/CD8 ratio decreased in the epidermis of elderly individuals, To analyze the changes of CD4 and CD8 cell numbers, the authors should examine the density or cell numbers of CD4 and CD8 in the epidermis.

→ We appreciate your important comments and added the results of immunohistochemical analysis on CD4 and CD8 T cells in the revised Figure 1A. Thanks to the reviewer's suggestion, we found that both CD4 and CD8 T cells increased in number/density in the skin of elderly subjects. We were not able to re-calculate the exact cell numbers from the flow cytometry results. On the bases of the results from immunohistochemistry and flow cytometry (CD4/CD8 ratio), we would like to presume that the number of both CD4 and CD8 T cells in skin, especially in epidermis, increase by aging, and the accumulating tendency is stronger in CD8 compared to CD4 T cells. We revised the description in the main text as follows (p.7, 1.4~7) :

'The density of T cells, both CD4 and CD8 fractions, increased in the epidermis of elderly individuals (Fig 1A). Since the CD4/CD8 ratio significantly decreased by aging (Fig 1B, S1B), it is suggested that CD8 T cells show the stronger tendency of being accumulated in the epidermis'.

2. The authors should examine T effector cells, T_{em} , T_{cm} along with T_{rm} to obtain a more comprehensive understanding of a dynamic change in populations of T cells with advancing age.

→ We appreciate your precious comment and totally admit the importance of including the information on the more detailed T cell phenotypes.

In order to clarify the aging dynamics of each T cell fraction in blood, we reanalyzed the existing data and added analyses on six new blood samples. In blood, after gating live CD4 and CD8 T cells, we defined $CD45RA^+CD45RO^-$ cells as naïve T cells, $CD45RA^-CD45RO^+CCR7^+CD62L^+$ cells as T_{CM} , $CD45RA^-CD45RO^+CCR7^+CD62L^-$ cells as T_{MM} , and $CD45RA^-CD45RO^+CCR7^-CD62L^-$ cells as T_{EM} . The decrease of naïve T cells and increase of T_{EM} was observed both in CD4 and CD8 fractions of elderly subjects (Supplementary Figure S1E).

As for skin T cell phenotypes, we reanalyzed the existing flow cytometry data in the original Supplementary Figure S1E. After gating live CD4 and CD8 T cells, we defined $CCR7^+CD62L^+$ cells as T_{CM} (original Figure S1E), $CCR7^+CD62L^-$ cells as

migratory memory T cells (T_{MM}), and $CCR7CD62L^-$ cells as T_{EM} . While the ratios of T_{CM} , T_{MM} , and T_{EM} in CD4 fraction did not significantly change by aging, T_{EM} increased in CD8 fraction along with the decrease of T_{CM} in elderly subjects. In this analysis, T_{EM} mostly included T_{RM} fraction as described previously (Yang Q et al, J Immunol, 2020).

We totally admit that this classification strategy is not thorough enough. We did not include CD45RO so we could not distinguish T_{EM} and T_{EMRA} in CD8 T cell fraction. However, since T cells in skin are recognized as having memory phenotype, we hope that these analyses would give us some information on the dynamic change in each T cell phenotype with advancing age. Unfortunately, we were not able to obtain additional skin specimens from benign tumor resection due to surgery restriction in the recent situation. If allowed, we would like to continue to accumulate data on each T cell fraction for more detailed understanding as a future project.

The results and gating strategies are shown in the revised Supplementary Figure S1E and Supplementary Figure S3D, E, respectively.

We revised the description in the main text as follows (p.7, 1.13~18).

‘Decline of blood naïve T cells was compensated by the expansion of effector memory T-cell (T_{EM}) population as previously reported (Fig S1E). On the other hand, in agreement with the increased frequency of skin T_{RM} which demonstrate T_{EM} -like phenotype, the proportion of recirculating T_{CM} in skin declined in CD8 fraction of older individuals (Fig S1E)’.

3. *IFN-g* and *TNF-a* production from CD8 T cells was significantly enhanced in the epidermis of elderly individuals in Fig. 1D, which may be due to the increased CD49a+ CD8 Trm cells (Fig. 1C), not the shift from TC17 to TC1.

→ We agree with your comment and admit that we cannot state if the same CD8 T cell could shift from TC17 to TC1 on the bases of our findings. We omitted the description of ‘shift’.

4. Fig. 2A shown that the production of *IFNγ* and *IL-17A* induced by pathogens (*S.au* and *C.alb*) was comparable in CD8 cells from young and elderly individuals, however, *IL-17A* production from CD8 T cells was significantly decreased in aged skin in Fig.1D. The authors should discuss it.

→ We apologize for the unclearness of the findings. We modified the description in the discussion part as follows in order to clarify the points we would like to state on the bases of the difference between unspecific stimulation and the stimulation induced by specific pathogens (p.11, 1.5~12).

‘Upon unspecific stimulation, *IL-17A* production from skin T cells was diminished in older individuals with an increased *IFNγ*-producing potential in the CD8 T-cell compartment after reversal of cell exhaustion. However, production of both *IL-17A* and

IFN γ in response to heat-killed *S.au* and *C.alb* was maintained in the skin T cells of elderly individuals. Thus, while skin T cells with IL-17A-producing potential generally decrease through aging, T cells reactive to specific pathogens which are frequent in skin are presumed to remain in skin sustaining their capacity of producing IL-17A and IFN γ through aging’.

5. In Fig. 4, skin T cells live longer than those in blood, what is the main mechanism for long-lived CD8 T cells in skin. The author should discuss it.

→ We appreciate your important comment. We presume that one of the main mechanisms for longevity of skin CD8 T cells is the microenvironment. We added the following description in discussion part (p.12, l.14~21).

‘Though the mechanism for regulating the maintenance of T_{RM} has not been well defined, one of the possible reasons for the longevity of T_{RM} may be microenvironments. The differentiation fate of T_{RM} depends on the microenvironment of the peripheral tissues, and other immune cells such as macrophages and dendritic cells provide niche factors to T_{RM} for their maintenance. The lipid-rich microenvironment of skin may also bring out the long-living potential of the adapted skin T_{RM} who are skillful at consuming fatty acids as the energy source in addition to glucose uptake’.

REVIEWERS' COMMENTS:

Reviewer #1 (Remarks to the Author):

None

Reviewer #2 (Remarks to the Author):

Dear authors, Thank you for your responses, which for me a perfectly suited for what was asked from me and the other reviewers. I have therefore no other comments, than to wish you good luck on your future work.

Reviewer #3 (Remarks to the Author):

The authors have addressed my concerns adequately, the revised manuscript has significantly improved.